# Direct Ink-Write Printing of Ceramic Clay with an Embedded Wireless Temperature and Relative Humidity Sensor

**DOI:** 10.3390/s23063352

**Published:** 2023-03-22

**Authors:** Cory Marquez, Jesus J. Mata, Anabel Renteria, Diego Gonzalez, Sofia Gabriela Gomez, Alexis Lopez, Annette N. Baca, Alan Nuñez, Md Sahid Hassan, Vincent Burke, Dina Perlasca, Yifeng Wang, Yongliang Xiong, Jessica N. Kruichak, David Espalin, Yirong Lin

**Affiliations:** 1Department of Mechanical Engineering, The University of Texas at El Paso, El Paso, TX 79968, USA; 2W.M. Keck Center for 3D Innovation, The University of Texas at El Paso, El Paso, TX 79968, USA; 3Sandia National Laboratory, Albuquerque, NM 87123, USA; 4Department of Computer Science Engineering, The University of Texas at El Paso, El Paso, TX 79968, USA; 5Department of Art, Ceramics Area, The University of Texas at El Paso, El Paso, TX 79968, USA

**Keywords:** Cone 5 porcelain clay, sensors, compression, rheology, X-ray diffraction

## Abstract

This research presents a simple method to additively manufacture Cone 5 porcelain clay ceramics by using the direct ink-write (DIW) printing technique. DIW has allowed the application of extruding highly viscous ceramic materials with relatively high-quality and good mechanical properties, which additionally allows a freedom of design and the capability of manufacturing complex geometrical shapes. Clay particles were mixed with deionized (DI) water at different ratios, where the most suitable composition for 3D printing was observed at a 1:5 w/c ratio (16.2 wt.%. of DI water). Differential geometrical designs were printed to demonstrate the printing capabilities of the paste. In addition, a clay structure was fabricated with an embedded wireless temperature and relative humidity (RH) sensor during the 3D printing process. The embedded sensor read up to 65% RH and temperatures of up to 85 °F from a maximum distance of 141.7 m. The structural integrity of the selected 3D printed geometries was confirmed through the compressive strength of fired and non-fired clay samples, with strengths of 70 MPa and 90 MPa, respectively. This research demonstrates the feasibility of using the DIW printing of porcelain clay with embedded sensors, with fully functional temperature- and humidity-sensing capabilities.

## 1. Introduction

Additive manufacturing (AM), also known as 3D printing, is a manufacturing technique that consists of a layer-by-layer deposition of materials to form a three-dimensional part from a CAD design. AM offers many advantages over traditional manufacturing due to its easier fabrication process, material availability, and relatively low manufacturing cost, making it a great tool for the rapid prototyping of complex geometrical designs. Although many advances have been achieved for metallic and polymeric materials for AM technology, there have been limited advances in the manufacturing of ceramics. The fabrication of ceramics by AM technology is an area of interest due to their excellent mechanical properties and thermal stability [1]. The most common AM methods to fabricate ceramics include direct ink-write (DIW) [2], selective laser sintering (SLS) [3], stereolithography (SLA) [4], binder jetting (BJ) [5], and material jetting (MJ) [6]. Printing techniques such as SLS and BJ allow the fabrication of highly complex geometrical shapes for ceramics. However, the printed parts usually result in a low density and poor mechanical properties [7]. On the other hand, printing methods such as SLA and MJ offer the possibility to fabricate high-density ceramics. Nonetheless, SLA and MJ result in a low material selectivity, restricting the fabrication process [8].

DIW is a material extrusion technique that consists of a layer-by-layer deposition of a paste through a nozzle by a pressure-driven mechanism onto a substrate. DIW offers a low-cost, relatively fast printing process and a large selection of printable materials with wide molecular weights that can easily be controlled by the rheological properties, making it suitable for prototyping. In addition, the DIW of ceramics usually does not require a heating temperature or photopolymerization process to retain its structural shape because the shear-thinning properties allow it to self-support during the printing process. The main drawback of using DIW to fabricate ceramics is that a post-treatment process is usually required to fully solidify the printed part. This post-treatment process (i.e., sintering or firing) allows for the removal of polymeric binders and moisture whilst enhancing the grain growth of ceramics, which contributes to an increase in its final density. However, this post-treatment process can also lead to different quality defects, including excessive shrinkage, voids, cracks, and warping [9]. Despite this, the biggest advantage of using the DIW technique to fabricate ceramics is that the final parts usually result in high-density samples with tunable mechanical properties [10,11].

Clay is one of the most widely available ceramics. Clay typically consists of a mixture of hydrated aluminosilicates with different ratios of silicon dioxide (SiO_2_) and aluminum oxide (Al_2_O_3_) [2]. Many research studies on clay printing have been used for decorative architectural [12], bioengineering [13], and construction [14] purposes. However, there is very limited information on clay printing applied for embedded sensors. Among the different types of clay, porcelain is of particular interest due to its mixture complexity (i.e., clay, kaolin, quartz, and fluxing agents) and its phase transformation during the sintering process that leads to an increase in its tangent loss at higher temperatures [15]. In addition, the rheological properties of porcelain can easily be controlled by mixing with water; thus, organic additives are not needed, which facilitates the firing process as debinding is not required [15].

Embedding sensors through additive manufacturing has grown over the years with the application of embedding sensing technology directly into manufactured parts to protect sensors from harsh environments as well as prolonging the lifetimes of sensors [16]. One main limitation and challenge with the additive manufacturing of embedded sensors field is the ability to create proper structures to fully enclose the sensor and maintain the integrity of the structure design [17]. The advantages of implementing embedded sensors with additive manufacturing allow a freedom of fabricating outstanding geometrical designs with the minimization of tooling for sensor enclosures [18]. Liu et al. investigated various forms of 3D printing with flexible strain sensors, testing the sensing mechanisms to view the feasibility of this method [19].

This work presents the development of a paste suitable for DIW printing using Cone 5 porcelain clay and deionized water. To demonstrate the printing capabilities of Cone 5 porcelain clay, different geometrical samples were fabricated and fired without producing cracks across the surface. In addition, the main purpose of this paste was to be able to embed a sensor during the printing process. The results obtained showed that was possible to fabricate embedded sensors capable of measuring temperature and relative humidity through wireless reading, proving the capability of using DIW printing for this application.

## 2. Experimental Details

### 2.1. Materials and Fabrication

Cone 5 porcelain (Armadillo Clay & Supplies, Austin, TX, USA) was selected as a solid loaded clay material (relative density: ~2.6 gm/cc; melting point: >1200 °C), and deionized (DI) water was used as a solvent for the slurry fabrication. A commercial S2 smart temperature and humidity sensor (MOAT^®^ TECHNOLOGIES LLC, San Diego, CA, USA) was used to measure the thermal and RH readings.

The slurry was fabricated by mixing the porcelain clay with DI water to control the viscosity and to allow for better flowability, printability, and shape retention. The clay slurry was hydrated to be viscous enough to flow through a feeding hose without clogging it whilst being capable of retaining its shape after deposition.

### 2.2. Printing and Post-Processing

Clay samples were printed using a Delta WASP 2040 Clay Printer (PicoSolutions Group Inc., Kearny, NJ, USA). The samples were printed using a nozzle diameter of 3 mm, print speed of 25 mm/s, layer height of 0.5 mm, and infill set to 30%. Compression disks were printed with dimensions of a diameter of 25 mm with a height of 3 mm. The sensor enclosure dimensions were 41 × 41 × 13 mm, as seen in Figure 1a, and dimensions of the non-fired parts were 40 × 40 × 12 mm, as seen in Figure 1b. Printed bisque parts were fired using a KILNMASTER LT (Skutt Ceramics Production Inc., Portland, OR, USA) at 1222.2 °C for six hours and then cooled down to room temperature. Figure 1 shows the porcelain clay samples: a clay non-fired sample (a), and fired sample with an applied Cone 6 glaze (1184–1222 °C) (b).

### 2.3. Material Characterization

Rotational rheological measurements for the clay slurry were conducted using a DHR-2 rheometer (TA Instruments, New Castle, DE, USA) with a parallel plate geometry. The test was performed with a 0.7 mm gap between the plates at 25 °C. The crystal structure was analyzed for the non-fired and fired parts by using X-ray diffraction (XRD), using CuKα radiation on a DISCOVER diffractometer (Bruker, Boston, MA, USA). The compression test was analyzed for the non-fired and fired parts by using an Instron 60TM-50 (Instron, Norwood, MA, USA) with a 50 kN force capacity.

## 3. Results and Discussions

### 3.1. Rheological Measurements

The measurement of the viscoelastic properties of clay slurry is a crucial stage in order to control the shape retention and printability [20]. Universal parameters for the creation of slurries do not exist as many variables affect the rheological behavior of the ceramic. An important factor in the creation of a slurry is the type of additive selected, such as polyvinyl alcohol, polyethylene glycol, and DI water, which are commonly used in the fabrication of ceramics using DIW [17,18]. Incorporating these types of additives allows for the material to flow better, with either Newtonian or non-Newtonian flow behaviors. When the stress is increased to a certain level, non-Newtonian fluids show signs of shear-thinning behavior such that their viscosity significantly declines as the shear rate rises [19]. Shear-thinning behavior is important to be able to extrude the clay without clogging the nozzle and subsequently achieving structural integrity after depositing the extrudate.

This behavior for non-Newtonian fluids (shear-thinning) can also be described by the Herschel–Bulkley model [20]: $\tau =\tau_{\gamma }+K\left(\tau \right){\dot{\gamma }}^{n}$. A high solid loading content is the most preferable parameter when designing a ceramic slurry. Increasing the solid loadings not only reduces the shrinkage in the final part, but also a high density can be achieved. Nevertheless, high contents of solid loadings drastically increase viscosity. The rheological data showed, when clay was combined with water at a specific wt.%, that the paste achieved an ideal printing viscosity for DIW of between $10^{3}\mathrm{and}\;10^{5}\;\mathrm{Pa}\cdot s$ [21]. Rheological data were taken for three different clay compositions, consisting of 15.0 wt.%, 16.2 wt.%, and 17.4 wt.% (i.e., water to clay), and can be seen in Figure 2. The clay slurry with 16.2 wt.% water (approximately 32 vol.% of clay) had an optimal shear-thinning behavior and achieved a well-dispersed slurry with this solid content, with no agglomerations or phase changing during the printing process, resulting in an even deposition of layers.

All slurries exhibited non-Newtonian behavior, yet small water percentage changes of ±1.2 wt.% were shown to significantly influence the viscosity as a function of the shear rate. Viscosity increased as the shear rate increased in a similar manner with the 15.0 wt.% compositions. A lower viscosity was noted for 17.4 wt.% whilst the viscosity for 16.2 wt.% was more stable, decaying linearly with a smaller slope and achieving a higher viscosity at higher shear rates, which would not pour beneath a particular yield stress [22,23]. The slurry reached an ideal volume fraction when the clay was at 34 vol.%, which was the calculated density of the printed part with the 16.2 wt.% ratio, making it suitable for DIW printing.

### 3.2. Clay 3D Printing Optimization

Printing parameters were established based on the clay slurry viscosity in order to obtain printed parts with different shapes. A nozzle with a diameter size of 1.5 mm was used to maintain a controlled material flow rate. The printing and travel speeds were set to 25 and 30 mm/s, respectively, to allow proper material flowability and reduce the material overflowing from the printed sides. The layer height was set to 0.8 mm with a line width of 1.5 mm to allow proper layer–layer adhesion. Figure 3 shows a printed part of a floater cup, which demonstrates the results of the fabrication of the materials and printing parameters. Figure 3a shows an initial design with minimal support around the handle, which caused an overhang to occur that nearly collapsed from the weight of the handle base. Figure 3b shows a redesigned model with a shorter and thicker handle with additional supports to prevent overhangs from occurring. Due to the weight of the moisturized clay, the use of further supports was required to retain the structural design of the printed parts with no failures.

### 3.3. Material Fabrication with an Embedded Sensor

A customized geometrical design was fabricated to print an enclosure to house a wireless sensor with temperature- and RH-reading capabilities. The design accounted for the shrinkage percentage of the clay during the drying process as well as the geometry shape of the sensor. The sensor was fully bound and the printed enclosure was left to dry overnight in a room temperature environment. The printed part exhibited minimal caving and deformation on the top surface of the sample and had no cracking during the drying process due to the controlled w/c of 16.2 wt.%. By achieving a proper viscosity of approximately 10^4^, the porcelain could maintain its shape whilst avoiding the top surface from caving in around the edges of the sensor, maintaining the intended structural design. The sensor was placed in the center of the printed enclosure mid-print by pausing the printing process and resuming once the sensor was placed, as seen in Figure 4A,B.

### 3.4. X-ray Diffraction Analysis

In Figure 5, the Cone 5 porcelain phase analysis is shown through an XRD analysis. A comparison between non-fired and fired samples was conducted to characterize the crystallinity phases of the samples after high-temperature post-processing. The analysis showed a decrease in crystallinity within the fired sample at 2θ at 25° and 27°, which indicated an increase in the glassy phase due to the melting of quartz whilst maintaining a constant mullite percentage [24]. Similar results have shown that an increase in temperature during post-processing (1300 °C) leads to a decrease in the peak intensity within the same angle range [25,26]. Jeoung-Ah showed similar results when running XRD on paper composite porcelain that was fired at different temperatures to examine the peaks, which showed a decrease in the ∝-quartz and mullite structure [27].

### 3.5. Compression Test

Non-fired and fired clay samples were tested under compression. The non-fired clay samples had a higher ultimate strain, demonstrating a more elastic behavior and achieving double the strain percentages than fired clay (1.0% vs. 2.0% at 70 MPa), as seen in Figure 6. The fired samples were denser due to grain growth during sintering and because the water was evaporated during the heat treatment process. The samples became brittle, thus reducing their elastic properties [28]. These results were consistent with studies that showed that clays that had moisture present or that had regained moisture after firing increased their expansive strain as well as their mass gain, mainly arising from the water content or rehydration processes [29]. Fired clay undergoes a steeper curve of stress vs. strain relation; it had less change in length when experimenting with the same loads over the same amounts of time as non-fired clay. Fired clay shows a more rigid behavior with sharp but short increments in strain as opposed to non-fired clay, which has a smoother curve with an exponential tendency. The same phenomena could be assumed for non-fired clay in this case study. The fired samples contained higher contents of water and mass after undergoing a sintering process that evaporated their water content [30].

### 3.6. Temperature and Relative Humidity Sensing

The sensing data at uncontrolled levels of temperature and humidity for the embedded sensor were simultaneously recorded. The MOAT sensor had a proven test range of 121 m, recording data every second; nevertheless, the wireless sensing capabilities of the embedded device were measured at a maximum distance of 141.7 m, meaning the clay enclosure did not have any impact on reducing the data transmission distances. The temperature reading showed it was capable of reading the maximum temperature of the sensor of 85 °F and RH of up to 40%. Figure 7 shows the temperature and RH readings with respect to time; both measurements were concurrently analyzed with the maximum distance determination.

The embedded sensor was placed under uncontrolled conditions in an outdoor environment for data acquisition to assess the capabilities of the embedded sensor in reading outdoor environment temperatures. It was observed that the embedded sensor had a delay of approximately 600 s (25 s delay when not embedded) before the external conditions penetrated the clay enclosure and the sensor could perceive the environmental conditions. The RH and temperature were correlated and showed the same peaks and downshifts during the same periods of time, most noticeable at 900 s and 1000 s.

The clay enclosure provided the sensor with protection from harsh environments, allowing constant monitoring. Although the thick porcelain barrier had a low transmissivity of environmental conditions, it only translated to a delay of 10 min (600 s) of data capturing, which outweighs the time and resources needed to take periodic measurements of typical harsh environments. An example is in the case of nuclear waste storage, which requires the periodic monitoring of tanks using Type B platinum–rhodium, 30% Rh–platinum rhodium, and 6% Rh thermocouplers; these allow measurements up to 1700 °C (very stable thermocouple, but less sensitive in the lower range) along with capacitive sensors to capture relative humidity data. Although the latter can last for long transient periods, they are characterized by large variations [31]. Our sensor had the capability to transmit data within the enclosure of the clay housing without disrupting the accuracy of the signal as it was within normal operating conditions (±0.3 °C/±0.5 °C for temperature and ±3.0% for RH), comparable with similar temperature sensors operating under ceramic enclosures [32].

## 4. Conclusions

In this study, Cone 5 porcelain clay was fabricated with an embedded wireless temperature and relative humidity sensor through DIW printing. Implementing DIW printing allowed a freedom of design in printing structures for the enclosure of the sensing unit. The established printing parameters of a printing speed of 25 mm/s, layer height of 0.5 mm, infill density of 30%, and nozzle diameter of 3 mm allowed the printing of the material with a proper flowability for proper layer–layer printing. This showed effectiveness with the printed designs that aided in avoiding structural deformation and minimal overhang from the layer–layer of the prints. The optimal water–clay mixing ratio of 1:5 allowed a proper printing process with good flowability of the porcelain clay whilst also allowing the parts to maintain their shape with minimal shrinkage to avoid cracking during drying. The compression strengths from the fired and non-fired samples were studied, showing strengths of 70 MPa for the fired sample and 90 MPa for the non-fired sample. An investigation into the crystallinity phase showed a small decrease within the mullite after the applied temperature during the firing process, which caused melting in the quartz, causing an increase in the glassy phase. The embedded sensor showed the capability of gathering readings whilst being enclosed within the clay at distances within 141.7 m from the reader. With clay particles, it allowed an increase in the electromagnetic behavior on the surface of the material, which allowed the frequency of the sensor to be able to be obtained for a thermal analysis. The results of sensing continuously showed temperature readings of 85 °F with humidity-sensing of up to 40% whilst embedded within the printed enclosure and being in an outdoor uncontrolled environment. The feasibility of this application allows usage as self-storage containers for hazardous materials or to be used in harsh environments without sacrificing the signal strength of the sensor within the enclosure. With Cone 5 porcelain, the corrosion resistance properties and great heat capacity make it a suitable insulating enclosure for these applications and uses.

## Figures and Tables

**Figure 1 sensors-23-03352-f001:**
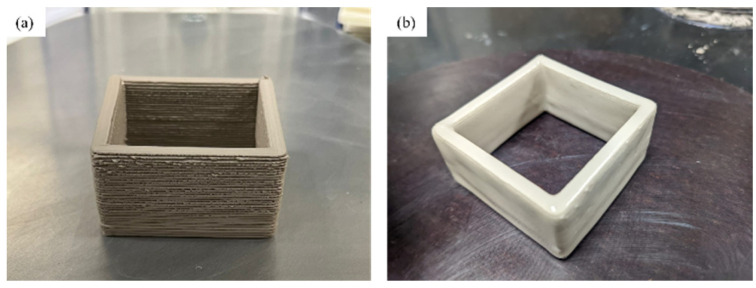
Cone 5 porcelain print through DIW: non-fired (**a**) and fired (**b**) samples.

**Figure 2 sensors-23-03352-f002:**
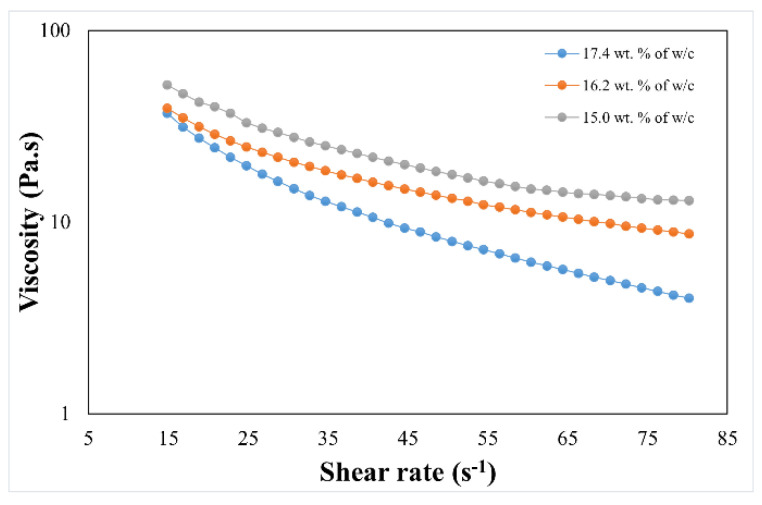
Data of viscosity as a function of shear rate for porcelain clay.

**Figure 3 sensors-23-03352-f003:**
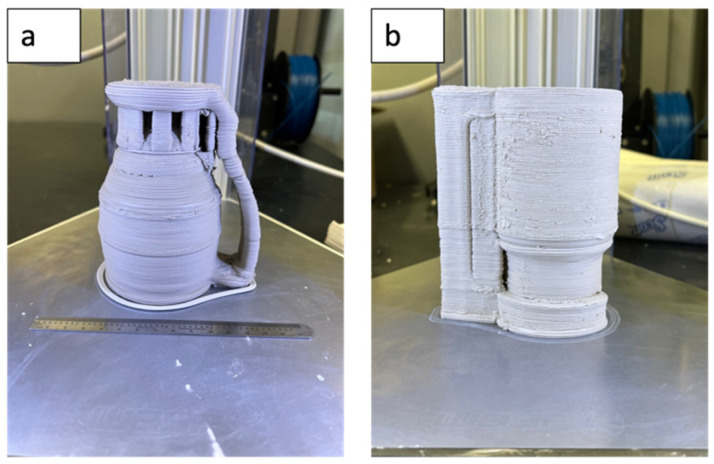
Initial (**a**) and redesigned (**b**) model design.

**Figure 4 sensors-23-03352-f004:**
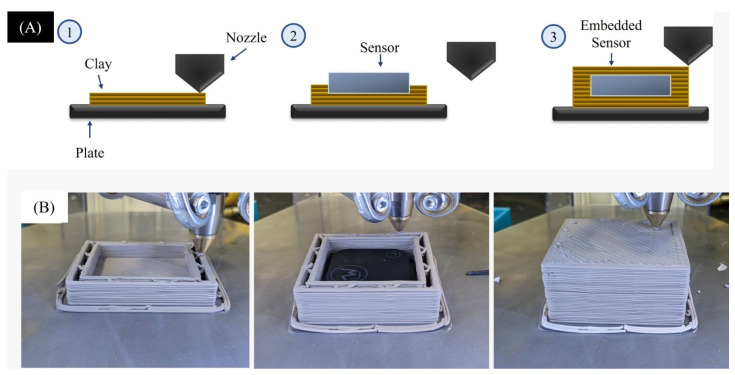
Schematic illustration of DIW printing with embedded sensor (**A**) and images of the embedded sensor printing stages (**B**).

**Figure 5 sensors-23-03352-f005:**
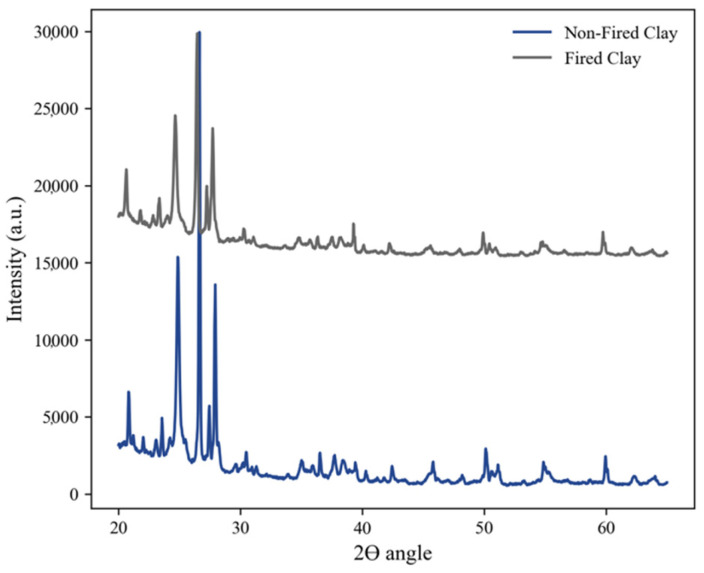
Porcelain clay XRD of non-fired and fired clay samples.

**Figure 6 sensors-23-03352-f006:**
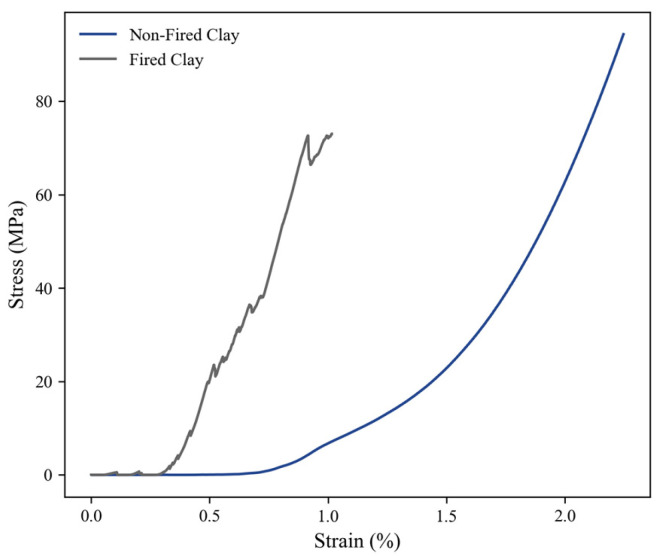
Stress vs. strain compression analysis of non-fired clay vs. fired clay.

**Figure 7 sensors-23-03352-f007:**
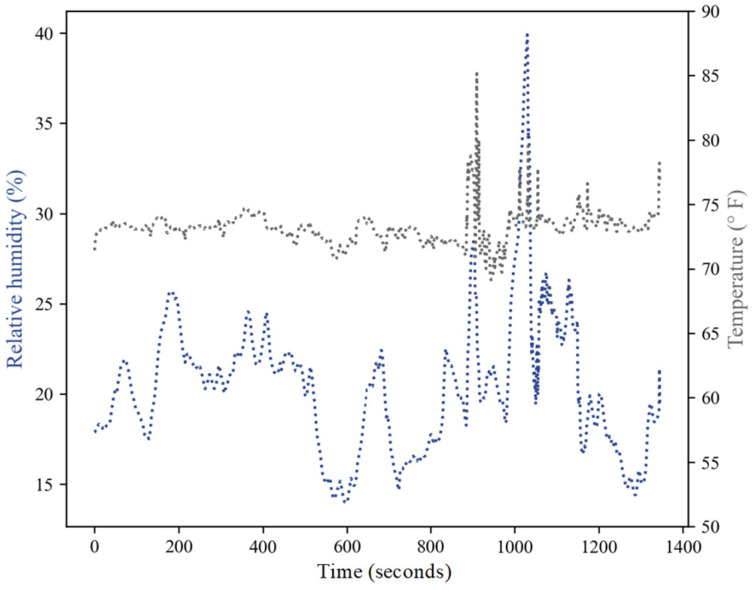
Relative humidity and temperature readings over time from the clay embedded sensor measured at 141.7 m.

## Data Availability

Noy applicable.

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
