# Peer review of "Direct Ink-Write Printing of Ceramic Clay with an Embedded Wireless Temperature and Relative Humidity Sensor"

_sensors, 2023, doi:10.3390/s23063352_

Round 1

Reviewer 1 Report

Point 1:  Introduction section must add the main scientific contribution, methods, main advantages and limitations of the proposed  method with respect to other similar methods reported in the literature.

Point 2: Make sure that always there is a space between the number and measurement unit in the whole document, ex. page 2. Also, minor spelling check is required.

Point 3: Page 6, Figure 6 is not mentioned in the text, but Figure 3.

Point 4: Introduce a picture for the structure having the sensor embedded More details about how the measurements were carried for the temperature and relative humidity sensing; a picture with the setup would be good.

Point 5: The comparison table with similar studies is missing.

Point 6: Which are the limitations or challenges of the proposed research?

Point 7: Conclusion section must be improved based on the above comments.

Reviewer 2 Report

- It's not clear what "32 vol.% of clay" mean. Is it the volume ratio of clay to the volume of water or the whole volume of the slurry? How the volume ratio was measured? what would be the volume ratio of clay for the 17.4% one? 

-  The authors mentioned the ideal fraction was at 34 vol.% and 27 vol.% display a non-newtonian behavior. Maybe the author should also study mixtures with a broader range of clay concentrations. For example 5 clays while including 34vo.% and 27vol.%. It was not clear how the authors selected these 2 concentrations. Maybe it's also an idea to only use volume concentration for consistency.

-why the viscosity of 16.2 wt.% water-clay is higher than that of 17.4 wt.% water-clay when the shear rate is higher than 2 or 3 /s? I assume the mixture containing more clay will always have a higher viscosity.

-Could the author add a paragraph in the introduction to introduce why embedded sensors are important and what are the applications?  What is the firing temperature will the embedded sensor be destroyed during the firing? Could the author provide the model number of the commercial sensor?

-The sentence at the end of page 6 and the beginning of page 7 is too long. Please write.

 - I am curious is there any specific reason the author choose this commercial sensor. Would it be possible to fabricate the sensors by DIW? 

Round 2

Reviewer 1 Report

Minor revision is required. 

Reviewer 2 Report

I have no further comment.